# Cerebellar and vestibular nuclear synapses in the inferior olive have distinct release kinetics and neurotransmitters

Josef Turecek, Wade G Regehr*

Department of Neurobiology, Harvard Medical School, Boston, United States

**Abstract** The inferior olive (IO) is composed of electrically-coupled neurons that make climbing fiber synapses onto Purkinje cells. Neurons in different IO subnuclei are inhibited by synapses with wide ranging release kinetics. Inhibition can be exclusively synchronous, asynchronous, or a mixture of both. Whether the same boutons, neurons or sources provide these kinetically distinct types of inhibition was not known. We find that in mice the deep cerebellar nuclei (DCN) and vestibular nuclei (VN) are two major sources of inhibition to the IO that are specialized to provide inhibitory input with distinct kinetics. DCN to IO synapses lack fast synaptotagmin isoforms, release neurotransmitter asynchronously, and are exclusively GABAergic. VN to IO synapses contain fast synaptotagmin isoforms, release neurotransmitter synchronously, and are mediated by combined GABAergic and glycinergic transmission. These findings indicate that VN and DCN inhibitory inputs to the IO are suited to control different aspects of IO activity.

*For correspondence:
wade_regehr@hms.harvard.edu

**Competing interests:** The authors declare that no competing interests exist.

## Introduction

At most synapses, neurotransmitter release is rapid, highly synchronous and tightly associated with presynaptic firing (*Borst and Sakmann, 1996*; *Katz and Miledi, 1965*; *Sabatini and Regehr, 1996*). At some specialized synapses release persists for tens or hundreds of milliseconds after a presynaptic action potential and is highly asynchronous (*Atluri and Regehr, 1998*; *Best and Regehr, 2009*; *Hefft and Jonas, 2005*; *Iremonger and Bains, 2007*; *Lu and Trussell, 2000*; *Peters et al., 2010*). Within the IO, the properties of neurotransmitter release at inhibitory synapses are highly variable between different subnuclei (*Best and Regehr, 2009*; *Turecek and Regehr, 2019*). In some IO subnuclei, GABA release is exclusively asynchronous, for others release is exclusively synchronous, and for the rest release has both a synchronous and an asynchronous component. This influences the jitter, the rise time and the decay time of synaptic responses (*Figure 1A*, *Turecek and Regehr, 2019*). Across the IO, the synchrony of GABA release is determined by the presence or absence of fast Synaptotagmin isoforms, Syt1 and Syt2 (Syt1/2). Release in the IO is synchronous if Syt1/2 are present, but asynchronous if they are absent (*Turecek and Regehr, 2019*), consistent with a role of Syt1/2 in release synchrony at other synapses (*Bacaj et al., 2013*; *Chen et al., 2017*; *DiAntonio and Schwarz, 1994*; *Geppert et al., 1994*; *Kochubey and Schneggenburger, 2011*; *Xu et al., 2007*).

The striking diversity in the properties of inhibitory synapses in the IO raises several important issues. The kinetics of release are determined by the presence of Syt1/2, but it is unclear whether the same presynaptic inputs provide both synchronous and asynchronous release. Neurons in the deep cerebellar nuclei (DCN) are a major source of inhibition (*Fredette and Mugnaini, 1991*), with DCN to IO inputs anatomically segregated such that a single DCN region projects to a single IO subnucleus (*Anguat and Cicirata, 1982*; *Dietrichs and Walberg, 1986*; *Dietrichs et al., 1985*; *Pijpers et al., 2005*; *Ruigrok, 1997*; *Ruigrok and Voogd, 1990*). It is possible that neurons in different DCN regions differentially express Syt1/2 to tune the kinetics of inhibition within each IO subnucleus. Some IO subnuclei also receive inhibitory input from vestibular nuclei (VN, *Barmack, 2006*;

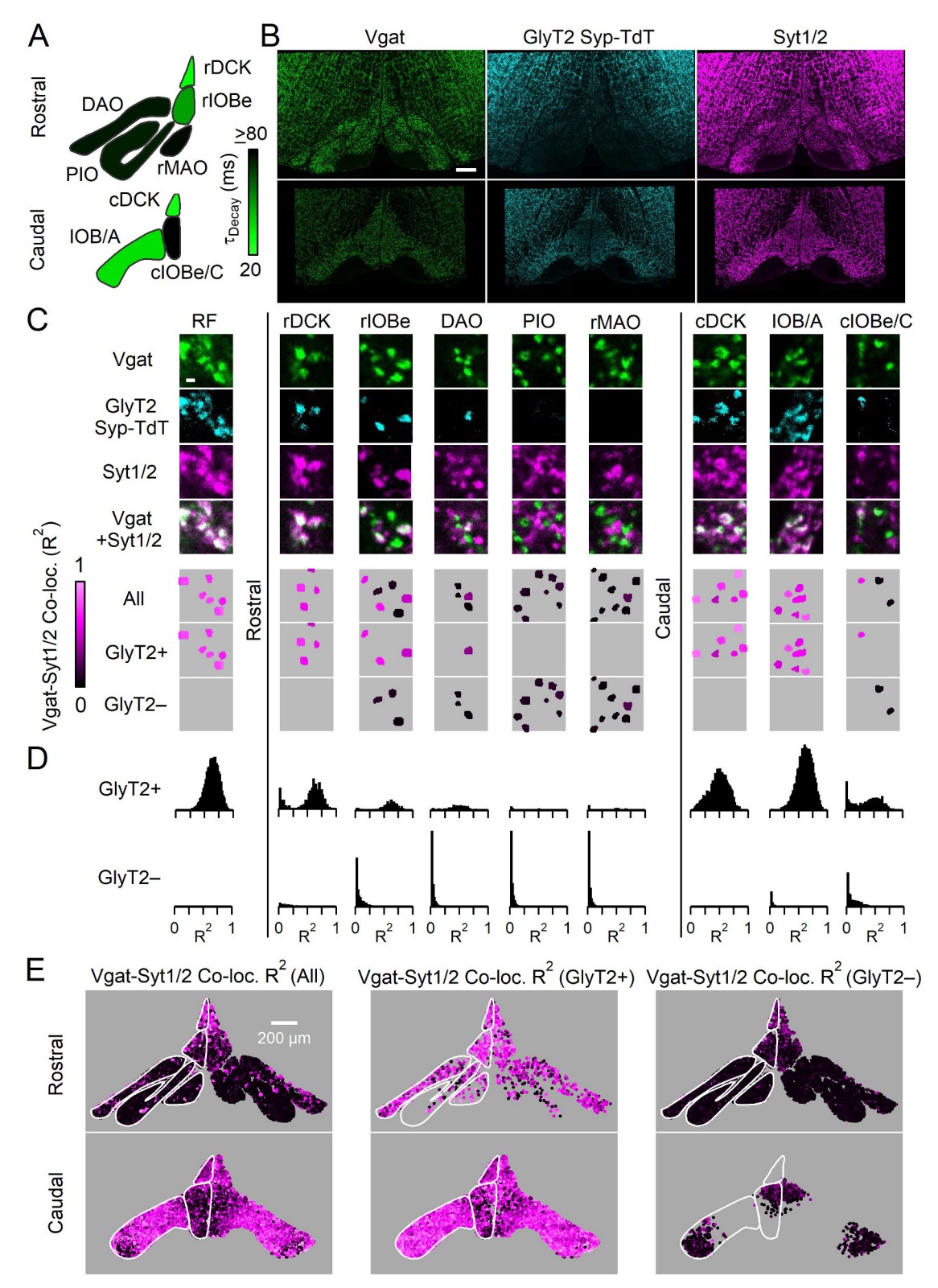

**Figure 1.** Syt1/2 is expressed at glycinergic synapses and absent from exclusively GABAergic inputs to the IO. (**A**) Summary of time course of the IPSC decay for the GABAergic component for different regions of the IO (from *Turecek and Regehr, 2019*). Abbreviations are defined in *Figure 7*. (**B**) Low-magnification of coronal sections of the IO immunolabeled for Vgat (green, left) to mark GABAergic and glycinergic inputs in a GlyT2$^{Cre}$; *R26$^{LSL-synaptophysin-TdT}$* (Ai34) mouse, Synaptophysin TdT labeled glycinergic boutons (cyan, middle), and immunostaining labeled Syt1/2 (magenta, right).

*Figure 1 continued on next page*

*Figure 1 continued*

Images are of the rostral (top) and caudal (bottom) IO. Scale bar is 200 µm. (**C**) High-magnification single plane images of inhibitory boutons from each subnucleus of the IO and the surrounding reticular formation (RF). From top to bottom: Vgat (green), Synaptophysin TdT in glycinergic boutons (cyan), Syt1/2 (purple), and a combined image of Vgat and Syt1/2. The lower three panels show analysis of each identified bouton. Boutons are color-coded for the degree of signal correlation of Vgat and Syt1/2 ($R^2$). Boutons are separated into Vgat-positive (top), glycinergic boutons (GlyT2+, middle), and non-glycinergic Vgat-positive boutons (GlyT2-, bottom). Scale bar is 1 µm. (**D**) Distribution of $R^2$ for all boutons in identified regions of the IO for glycinergic (top) and GABAergic synapses (bottom). (**E**) Signal correlation ($R^2$) of Vgat and Syt1/2 for all identified Vgat-positive boutons (left), for glycinergic boutons expressing TdT and Vgat (middle), and for exclusively GABAergic synapses expressing Vgat but not TdT (right).

The online version of this article includes the following source data and figure supplement(s) for figure 1:

**Source data 1.** Syt1/2 is expressed at glycinergic synapses and absent from exclusively GABAergic inputs to the IO.
**Figure supplement 1.** The presence of Syt1/2 at all Vgat-positive boutons (TdT+ and TdT−) in the ventral brainstem.
**Figure supplement 2.** Distribution of GlyT2 and Syt1/2 in Vgat-positive boutons in IO subnuclei.

*Barmack et al., 1998*; *De Zeeuw et al., 1993*; *Nelson, 1988*), but it is unknown whether VN inputs are synchronous, asynchronous or both. Another aspect of inhibitory transmission in the IO that is poorly understood is the contribution of glycinergic transmission to synchronous and asynchronous inhibition. In other brain regions GABA and glycine are often co-released, and glycinergic transmission often has faster kinetics than GABAergic transmission (*Apostolides and Trussell, 2013*; *Jonas et al., 1998*; *Lu et al., 2008*; *Moore and Trussell, 2017*; *O'Brien and Berger, 1999*). This raises the possibility that glycinergic transmission could contribute to fast synaptic inhibition in the IO, but the properties of glycinergic transmission are not known because previous studies have focused primarily on GABAergic inhibition in the IO.

We examined the properties of inhibitory synapses within the IO. Our anatomical and electrophysiological experiments indicate that many inhibitory boutons in the IO have a glycinergic component. Viral expression of ChR2-YFP allowed us to isolate the properties of DCN and VN inputs. We find that inhibitory inputs to the IO from the DCN are exclusively asynchronous and mediated by GABA$_A$ receptors. VN inputs provide rapid inhibition with synchronous release that activates GABA$_A$ receptors and glycine receptors. IO subnuclei with exclusively asynchronous release only received input from the DCN, regions with exclusively synchronous release received strong input from the VN, and regions with mixed kinetics received prominent input from both. These findings establish that different sources provide inhibitory synapses with distinct kinetics within the IO.

## Results

We used immunohistochemistry to characterize inhibitory synapses in the IO. We immunolabeled Vgat in animals expressing Synaptophysin-Tdtomato (Syp-TdT) driven by GlyT2-Cre (*Kakizaki et al., 2017*), and co-stained for Syt1/2 (*Figure 1B*). This strategy labels all inhibitory boutons with Vgat, and prominent labeling is present throughout the rostral and caudal IO and in regions surrounding the IO. Glycinergic boutons are present at modest levels in some regions of the rostral IO, and they are present at higher levels in the caudal IO and in the reticular formation (RF) surrounding the IO. Syt1/2 labeling is prominent throughout the IO, but Syt1/2 labeling is not restricted to inhibitory synapses, so much of this labeling is in glutamatergic synapses (*Turecek and Regehr, 2019*). We therefore identified Vgat-positive boutons and measured co-localization with Syt1/2 by correlating the signals of Vgat and Syt1/2. Signal correlation for each bouton was measured using Pearson's correlation coefficient ($R^2$), with low $R^2$ values indicating that Syt1/2 is absent from the bouton, and a large $R^2$ value indicating that Syt1/2 is present (*Turecek and Regehr, 2019*, *Figure 1—figure supplement 1*, *Figure 1—source data*, see Materials and methods). We also identified Vgat-positive boutons that are either glycinergic (TdT−containing, GlyT2+) or purely GABAergic (TdT−lacking, GlyT2−, *Figure 1C*). Differences in the properties of GlyT2+ synapses were evident in high power images of different subnuclei (*Figure 1C*), and in summaries of $R^2$ for all Vgat boutons in IO subnuclei (*Figure 1D*). Throughout the IO, Syt1/2 was absent in GlyT2− boutons, whereas Syt1/2 was present at most GlyT2+ synapses. GlyT2+ synapses were also segregated to distinct IO subnuclei. In the rostral IO, a majority of inhibitory synapses lacked Syt1/2 and were GlyT2− in the PIO and rMAO (*Figure 1—figure supplement 2*). Other subnuclei contained mixed populations of GlyT2+, Syt1/2-expressing synapses (DAO, rIOBe/C, cIOBe/C). In the caudal IO and dorsal cap of Kooy, most

synapses were GlyT2+ and expressed Syt1/2 (rDCK, cDCK, IOB/A). The anatomical segregation of GlyT2+ synapses was evident in the distributions (*Figure 1D*, *Figure 1—figure supplement 2*, *Figure 1—source data 1*), and reconstructed maps of GlyT2+ and GlyT2− boutons in the IO (*Figure 1E*). Our results indicate that glycinergic inputs to the IO are generally directed to distinct subnuclei and mostly express Syt1/2, whereas exclusively GABAergic inputs are targeted to complementary nuclei and lack fast Syt isoforms.

Nothing is known about the function of glycinergic transmission in the IO because previous studies of inhibition in the IO focused on GABAergic transmission and were therefore conducted in the presence of glycine receptor antagonists (*Best and Regehr, 2009*; *Turecek and Regehr, 2019*). However, our immunohistochemical studies make several predictions about inhibitory synapses in the IO. They suggest that glycinergic synapses are rapid due to the presence of Syt1/2. In contrast, exclusively GABAergic synapses are expected to be asynchronous based on the lack of Syt1/2. In addition, the spatial dependence of glycinergic boutons provides a guide to subregions where glycinergic synaptic responses are likely to be prominent. We performed electrophysiological studies to test these predictions and to characterize the glycinergic inputs to the IO.

The DAO is a particularly interesting place to test the hypothesis that glycinergic synapses are synchronous. In the DAO, GABAergic transmission is dominated by asynchronous release, and the slow average synaptic currents reflect variable long-latency occurrence of individual quantal events on individual trials (*Best and Regehr, 2009*). We also found that in the DAO, GABA is primarily released asynchronously, and that most inhibitory synapses lack Syt1/2 (*Turecek and Regehr, 2019*). However, a small subset of inhibitory synapses in the DAO contain both Syt1/2 and glycinergic markers (*Figure 1C–E*; *Figure 1—figure supplement 2*). We therefore re-examined the properties of evoked inhibitory transmission in the DAO with glycinergic and GABAergic transmission intact, and with excitatory input blocked by NBQX and (R,S)-CPP. These experiments, and most others in this study, were performed in conditions suited to measuring the kinetics of release (single stimuli, 2.5 mM external calcium). Experiments were performed at room temperature to avoid the subthreshold oscillations that are prominent at physiological temperatures and that make studies of synaptic transmission difficult. Inhibitory synaptic responses had two components of release with very different properties: a rapid synchronous component consistent across trials (*Figure 2A*, left), was followed by a prolonged asynchronous component that varied across trials as described for asynchronous release of GABA in the DAO. Gabazine eliminated asynchronous release and left a rapid synchronous component intact (*Figure 2A*, right). We plotted the evoked currents sensitive to gabazine and strychnine that reflect the properties of GABA and glycine release respectively (*Figure 2B*). Compared to GABA, the release of glycine had much faster rise and decay kinetics and had similar properties to synchronous release seen in synapses outside the IO (*Figure 2A–C*). We observed considerable variability in the amplitude of the glycinergic component of release for different cells in the DAO, but GABA release typically contributed much more overall inhibitory charge transfer (*Figure 2C*, *Figure 2—source data 1*). Thus, in agreement with the predictions from immunohistochemistry, the DAO receives two distinct forms of inhibition. The synchronous component has a prominent glycinergic component, whereas the asynchronous component is exclusively GABAergic.

We also measured the properties of mIPSCs to see if regions in which immunohistochemically identified glycinergic boutons are abundant also have prominent glycinergic mIPSCs. We recorded mIPSCs in the presence of either gabazine to isolate glycinergic events, or strychnine to isolate GABAergic events. We found that the frequency of glycinergic mIPSCs is regionally segregated. In the PIO where mIPSCs mediated by GABA are abundant, we did not observe spontaneous glycine release (*Figure 2D*). In the caudal A and B subnuclei of the IO (IOB/A), we observed prominent glycinergic mIPSCs (*Figure 2E*). Glycinergic mIPSCs across all subnuclei of the IO had more rapid kinetics and larger amplitudes than GABAergic mIPSCs (*Figure 2F,G*, *Figure 2—source data 1*). The fast decay time of individual glycinergic quantal events contributes to the rapid kinetics of evoked glycine release we observed in the DAO. mIPSCs measured in the absence of antagonists of GABA$_A$ and glycine receptors revealed large amplitude events with a slow decay that appeared to consist of both glycinergic and GABAergic components (*Figure 2—figure supplement 1*, *Figure 2—source data 1*). Washing on gabazine eliminated small and slow mIPSCs, whereas large amplitude mIPSCs remained. The decay of the remaining large mIPSCs was faster in the presence of gabazine, suggesting that some glycinergic synapses in the IO also corelease GABA.

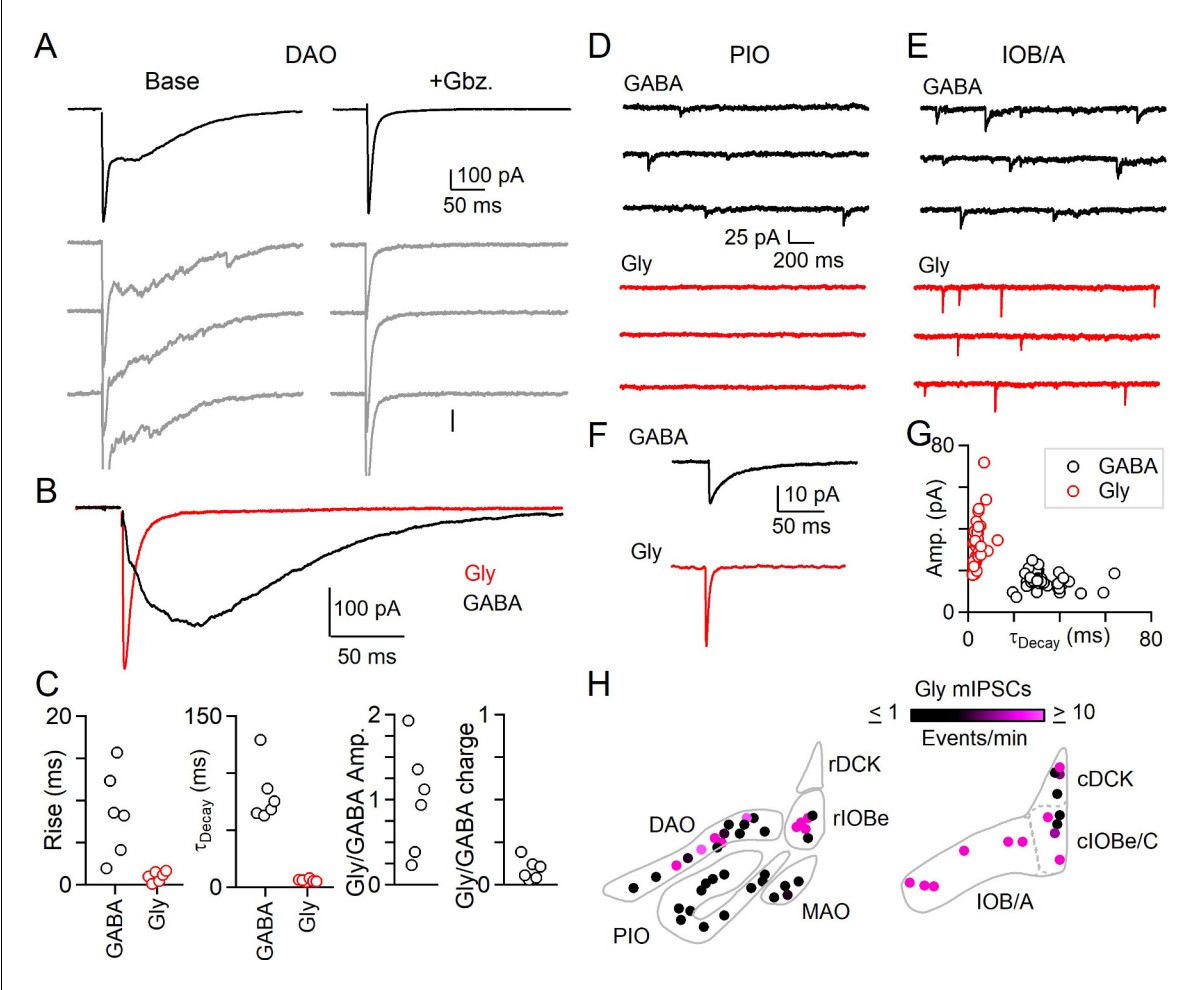

**Figure 2.** Evoked and spontaneous glycinergic input to the IO. (**A**) Electrically evoked IPSCs in the DAO with average (top, black) and individual trials (gray), before (left) and after washing of Gbz. (right). (**B**) Average Gbz.-sensitive IPSC (GABA, black) and strychnine-sensitive IPSC (Gly, red) for cell in A. (**C**) Properties of GABA and glycine components of evoked release in the DAO, showing the rise and decay kinetics (left) and ratios of glycine and GABA evoked amplitude and charge (right). Markers are individual cells. (**D**) Top: Example of mIPSCs in a PIO neuron recorded in the presence of strychnine (left). Bottom: Same as in top, but for a different PIO neuron recorded in the presence of Gbz. No events were detected over the course of 10 min. (**E**) Same as in D, but for two cells in the IOB/A subnuclei. F: Average mIPSCs for cells in E. (**G**) Plot of amplitude vs. decay time for averaged mIPSCs of all cells across the IO. Each marker is a cell. (**H**) Color-coded map of glycinergic mIPSC frequencies for all recorded cells in the rostral (left) and caudal (right) IO. Markers are individual cells.

The online version of this article includes the following source data and figure supplement(s) for figure 2:

**Source data 1.** Evoked and spontaneous glycinergic input to the IO.

**Figure supplement 1.** GABA and glycine can be co-released in the IO.

The frequency of glycinergic mIPSCs in the IO (*Figure 2H*) conformed to the observed prevalence of anatomically defined glycinergic synapses within subnuclei (*Figure 1*). Glycinergic mIPSCs were not observed in the PIO, and rMAO, where glycinergic boutons are not observed. At the opposite extreme, the frequency of glycinergic mIPSCs was reliably high throughout the IOB/A and other parts of the caudal IO, regions with a high density of glycinergic boutons. For regions such as the DAO and rIOBe where a subset of inhibitory synapses is glycinergic, glycinergic mIPSCs were apparent in a fraction of cells. The average frequency of glycinergic mIPSCs in these areas also was lower than for the IOB/A. Thus, both anatomical and electrophysiological approaches indicate that the contribution of glycinergic transmission to inhibition is highly subnucleus dependent.

Multiple regions provide inhibitory input to the IO, including the deep cerebellar nuclei (DCN) and the vestibular nuclei (VN) (*Figure 3A*). It is not clear whether a single source of inhibition can

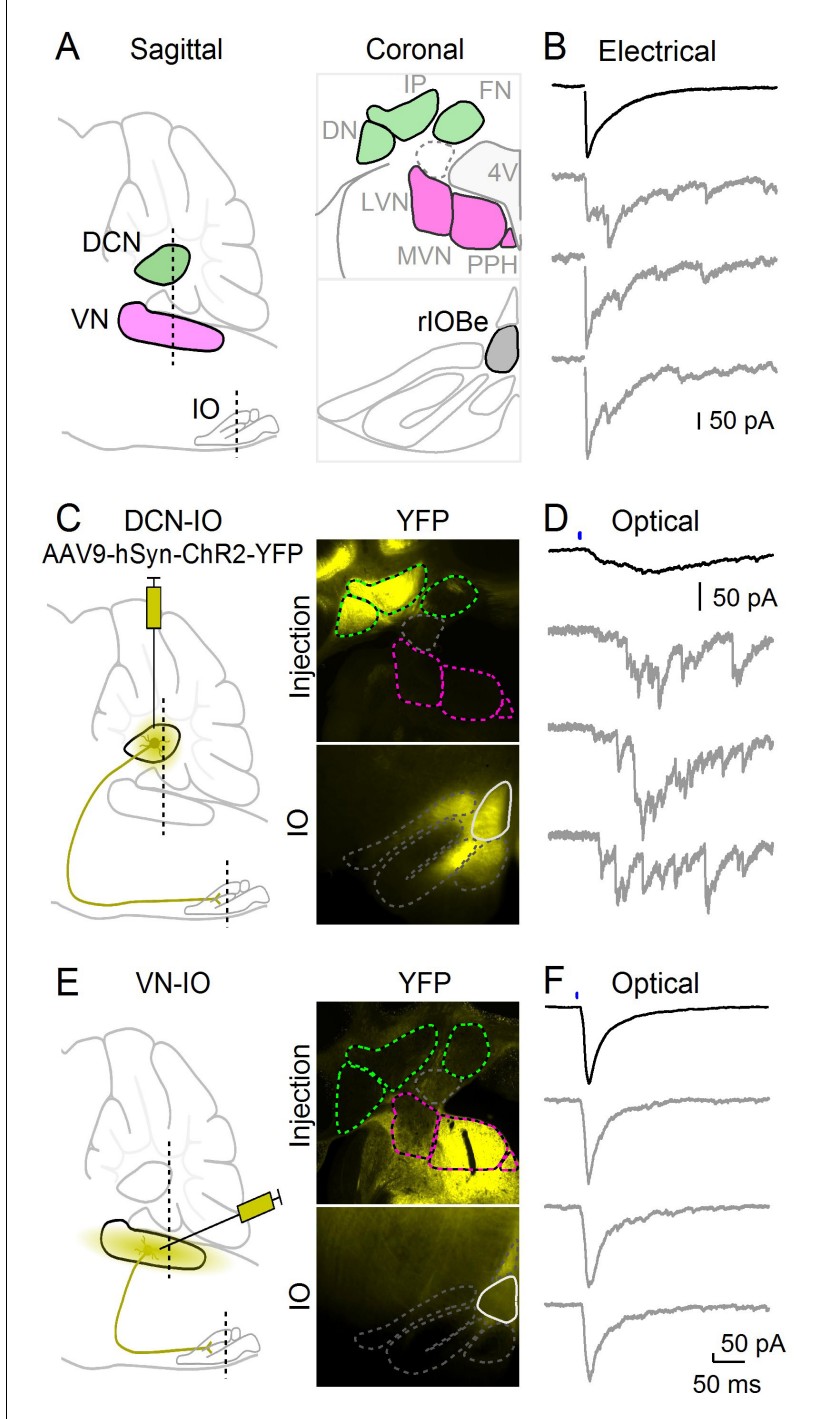

**Figure 3.** Optical isolation of cerebellar and vestibular sources of inhibition to the IO. (**A**) Sagittal view of the cerebellum and brainstem showing the locations of the DCN, VN and IO (left). Dashed lines indicate the coronal planes of insets showing the DCN and VN (top right) and IO (bottom right). Abbreviations are defined in *Figure 7*. (**B**) Electrically evoked IPSCs showing mixed synchronous and asynchronous release from a cell in the rIOBe, with average IPSC (black, top) and individual trials (gray, bottom). (**C**) Sagittal view of strategy for expressing ChR2-YFP in the DCN (left), with example injection site expressing ChR2-YFP in the DCN but not VN in coronal sections (top right). YFP-labeled projections are visible in the IO. (**D**) Optically-evoked IPSCs in the rIOBe from DCN injection shown in C. (**E**) Injection strategy for labeling of the VN (left). The injection pipette was angled to avoid the DCN. Example injection of the VN (top right) produced labeling in the IO and surrounding brainstem (bottom right). F: Optically-evoked IPSCs in the rIOBe from VN injection shown in E.

mediate both synchronous and asynchronous release, or whether different sources provide inhibition with distinct release kinetics. The beta subnucleus (rIOBe) is an ideal region to distinguish between these possibilities, because electrical stimulation evokes inhibition within the rIOBe with prominent synchronous and asynchronous components (*Figure 3B*). We assessed the properties of synapses provided by different sources by locally injecting AAV to restrict the expression of ChR2-YFP to either the DCN or the VN, and then measuring light-evoked responses in the rIOBe subnucleus. Injections into the DCN labeled fibers that projected broadly to the IO, including the rIOBe (*Figure 3C*). Light-evoked responses were entirely asynchronous and lacked a synchronous component (*Figure 3D*). The rise and decay times of the average IPSC were very slow, and there was considerable jitter in the peak times of IPSCs for individual trials. Injections into the VN also labeled fibers that projected to the IO including the rIOBe (*Figure 3E*), but light-evoked responses were entirely synchronous and lacked an asynchronous component (*Figure 3F*). The rise and decay times were fast, and single trial responses were very reproducible, with low jitter in peak times. Thus, the DCN and the VN both project to the rIOBe, but the properties of the synapses are very different.

This optogenetic approach was used to characterize DCN and VN projections to numerous IO subnuclei (*Figure 4*). DCN injections resulted in ChR2-YFP fluorescence throughout much of the IO, including the rIOBe, DAO, PIO, and the rMAO in the rostral IO, and in the lateral IOB/A and IOBe/C in the caudal IO. Example light-evoked responses are shown for DCN projections to the DAO, IOB/A and the PIO (*Figure 4A*). Light-evoked synaptic responses were observed in all regions where fluorescent fiber labeling was present (*Figure 4C*, filled circles), with the exception of the cIOBe/C (*Figure 4C*, open circles). The lack of evoked responses in the cIOBe/C suggests that the presence of ChR2-YFP fluorescence might be associated with DCN fibers of passage rather than synaptic boutons, although it is also possible that our injections did not label the region of the DCN that projects to the cIOBe/C. For all regions where DCN responses could be evoked, synaptic inputs had the slow rise, prolonged decay and large jitter typical of asynchronous release, as exemplified by the DCN inputs to the rIOBe (*Figure 4A,C,E*, *Figure 4—source data 1*). The properties of release are consistent with previous results showing that delayed release in the IO is composed of individual asynchronous quantal events (*Best and Regehr, 2009*). VN injections resulted in ChR2-YFP fluorescence in the rIOBe and DAO in the rostral IO, and in the IOB/A and IOBe/C in the caudal IO. Example light-evoked responses are shown for the DAO, IOB/A, and the cIOBe/C (*Figure 4B*). Light-evoked synaptic responses had prominent synchronous release with rapid rise times and decay times, and the peak of the synaptic responses had little jitter, as illustrated by VN inputs to the rIOBe (*Figure 4B,D, F*, *Figure 4—source data 1*).

Light-evoked responses from the DCN were exclusively asynchronous across subnuclei of the IO and we next asked whether boutons originating from the DCN lack Syt1/2. Viral expression of ChR2-YFP in the DCN allowed us to identify axons originating from the DCN (*Figure 4—figure supplement 1*). In the DAO where synchronous and asynchronous release occur (*Figure 2*), a subset of Vgat boutons were labeled with YFP. Almost all YFP-containing inhibitory boutons lacked Sy1/2 (362/371 boutons). We also examined boutons in the DAO where Syt1/2 were present and found that very few Syt1/2-containing inhibitory boutons were YFP-positive (10/345 boutons, *Figure 4— figure supplement 1*, *Figure 4—source data 1*). For the DAO, we find that synapses originating from the DCN are exclusively asynchronous and mostly lack Syt1/2.

DCN and VN synaptic inputs also differed in their pharmacological sensitivity (*Figure 5*). The asynchronous synaptic currents of the DCN were eliminated by gabazine, indicating that they were mediated entirely by activation of GABA$_A$ receptors (*Figure 5A–C*, *Figure 5—source data 1*). In contrast, gabazine did not eliminate the rapid synaptic currents mediated by VN inputs (*Figure 5D–F*). A very rapid component of the synaptic current remained in the presence of gabazine (*Figure 5D–F*, *Figure 5—source data 1*) and the remaining release could be eliminated by strychnine (not shown). The kinetics of the glycinergic component were much faster than those of GABA, consistent with the faster rise and decay times of individual glycinergic mIPSCs in the IO. Thus, VN inputs are mediated by synchronous release consisting of a very fast component mediated by glycine receptors, and a slower component mediated by GABA$_A$ receptors.

We next examined the properties of putative DCN and VN inputs to the IO under near-physiological conditions and more relevant firing patterns. Neurons in the IO are extensively electrically-coupled, and measurements of inhibitory input can be contaminated by currents spontaneously generated by electrically-coupled neighboring cells (*Figure 6—figure supplement 1*, *Figure 6—*

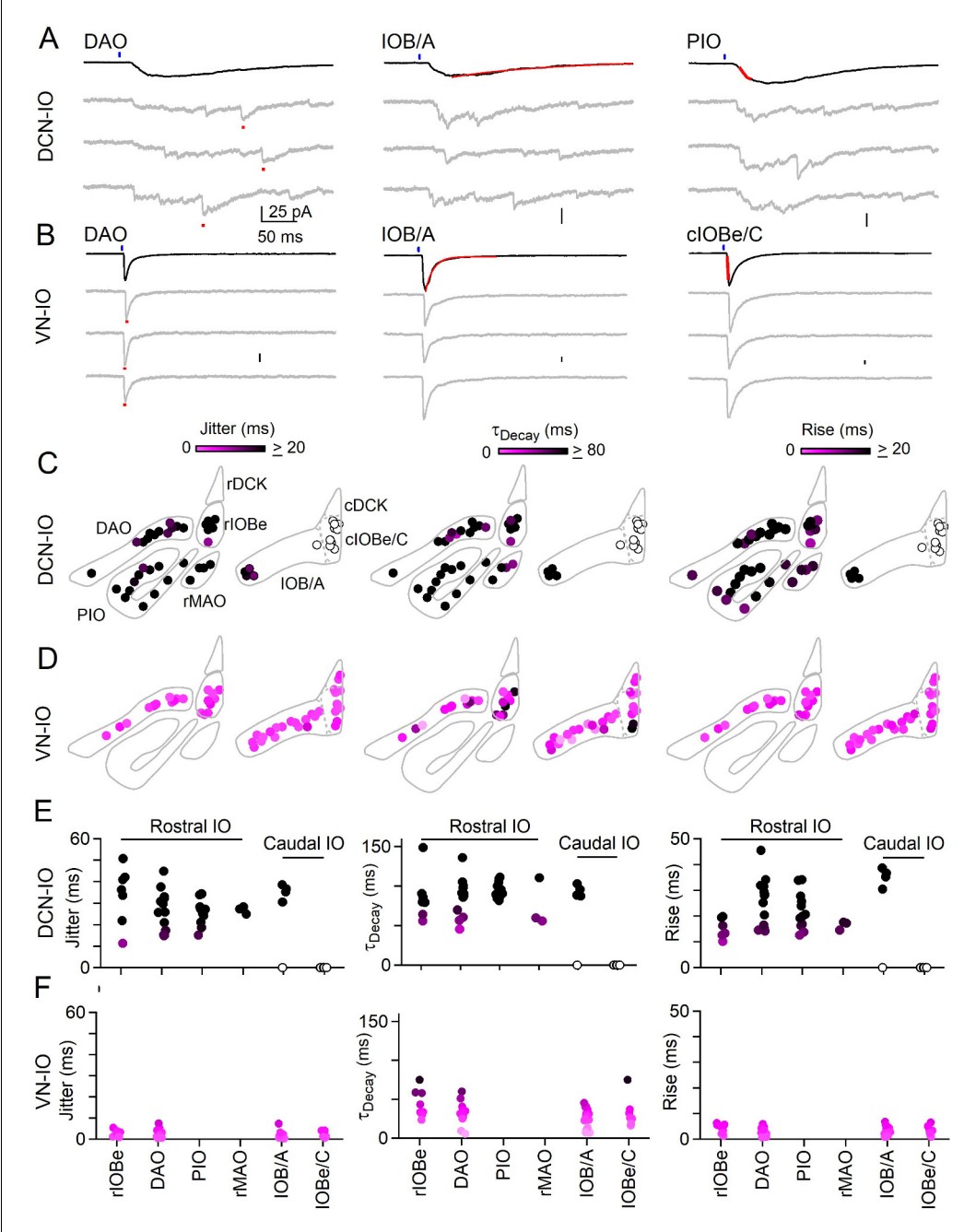

**Figure 4.** Synchronous and asynchronous inhibition arise from distinct presynaptic sources. (A) Light-evoked response for DCN inputs to the indicated regions with average responses (black) and individual trials (gray). Measurements of jitter, decay and rise time shown in red. (B) Same as A but for VN inputs. (C) Spatial plots of the jitter (left), decay time (middle) and rise time (right) of optically-evoked inhibitory input from the DCN for all recorded IO neurons. (D) Same as C but for inputs from VN. (E) Summary of jitter (left), decay time (middle) and rise time (right) for each region for inputs from the DCN. (F) Same as E but for inputs from the VN.

The online version of this article includes the following source data and figure supplement(s) for figure 4:

**Source data 1.** Synchronous and asynchronous inhibition arise from distinct presynaptic sources.

**Figure supplement 1.** Boutons originating from the DCN lack Syt1/2.

**Figure supplement 1—source data 1.**

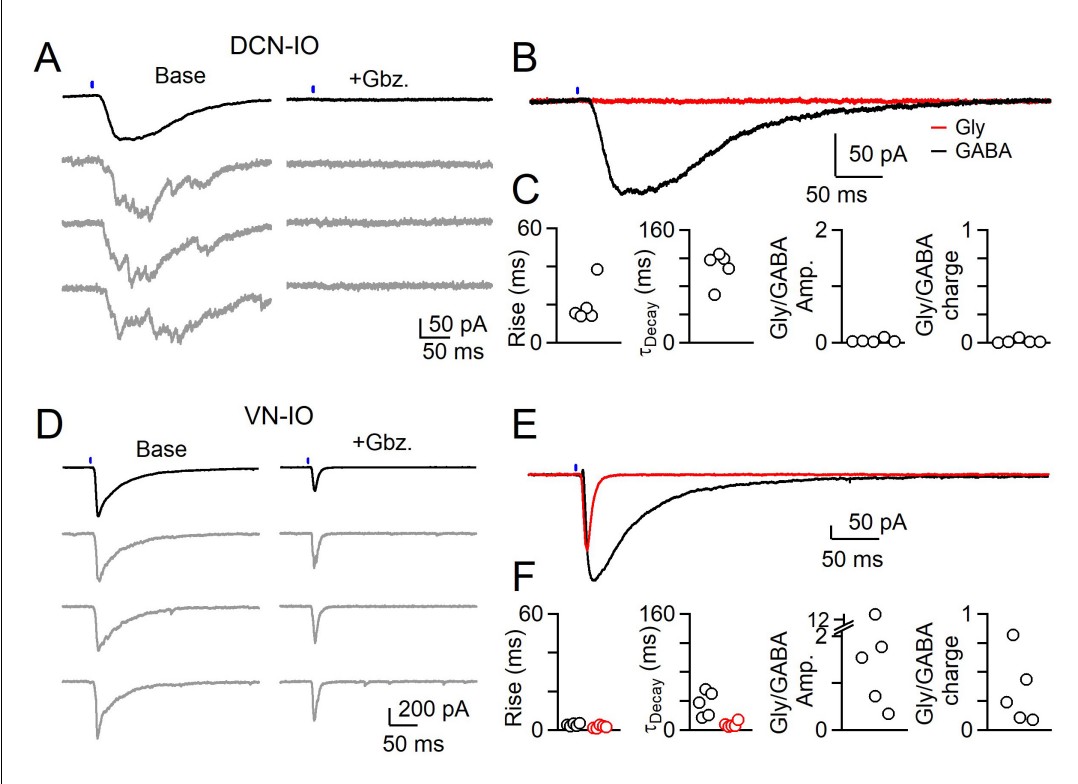

**Figure 5.** Different sources of inhibition to the IO use different neurotransmitters. (**A**) IPSCs evoked by optical stimulation of DCN inputs to a DAO neuron with average (black) and individual trials (gray) before (left) and after washing of Gabazine (left). (**B**) Average overlaid Gabazine-sensitive (black, GABA) and strychnine-sensitive (red, Gly) IPSCs for cell shown in A. (**C**) Properties of averaged GABA (black) and Glycine (red) components of release originating from the DCN for each cell, with rise (left), decay (middle) times and ratio of Glycine and GABA IPSC charge. No Glycine component was detected in any cell. (**D**) Same as in A, but for optically-evoked vestibular input to a cell in the IOB/A. (**E**) Overlaid averaged Gabazine-sensitive (black, GABA) and strychnine-sensitive (red, Gly) IPSCs for cell shown in D. (**F**) Same as C, but for properties of vestibular inputs to IO neurons.

The online version of this article includes the following source data for figure 5:

**Source data 1.** Different sources of inhibition to the IO use different neurotransmitters.

*source data 1*). We performed the bulk of our experiments under conditions that maximize synaptic release and minimize bursting and subthreshold oscillations from neighboring electrically-coupled neurons (see methods). The properties of synapses can be very sensitive to experimental conditions, and we therefore measured release under more physiological temperatures (35°C) and external $Ca^{2+}$ (1.5 mM).

Differences in the properties of inhibition between IO subnuclei were persistent in more physiological conditions. We first focused on the IOB/A, where we found that inhibition is synchronous, mediated by GABA and glycine, and primarily originates from the VN. Under near-physiological conditions we found that inhibition in the IOB/A was synchronous, but had more rapid kinetics than at room temperature (*Figure 6A*). We next examine the PIO, where we found that inhibition is exclusively asynchronous, mediated by GABA and primarily originates from the DCN. Under near-physiological conditions, release in the PIO was also asynchronous (*Figure 6B*), but was generally more difficult to evoke with single stimuli. Comparing the IOB/A and PIO, we found that release in the IOB/A was tightly synchronized with very little jitter and rapid decay and rise kinetics, whereas release in the PIO remained exclusively asynchronous with much slower decay and rise kinetics (*Figure 6C*, *Figure 6—source data 1*).

Neurons of the DCN and VN are spontaneously active in vivo, and we therefore next compared inhibition in the IOB/A and PIO using trains of stimuli. We used electrical stimuli because optical stimuli can be unreliable at recruiting axons at high stimulation frequencies. A 50 Hz train of 10 stimuli evoked rapid and phasic inhibition in the IOB/A. *Figure 6D* shows two example cells from the IOB/A that received synchronous inhibitory input. Between stimuli, IPSCs decayed rapidly and there

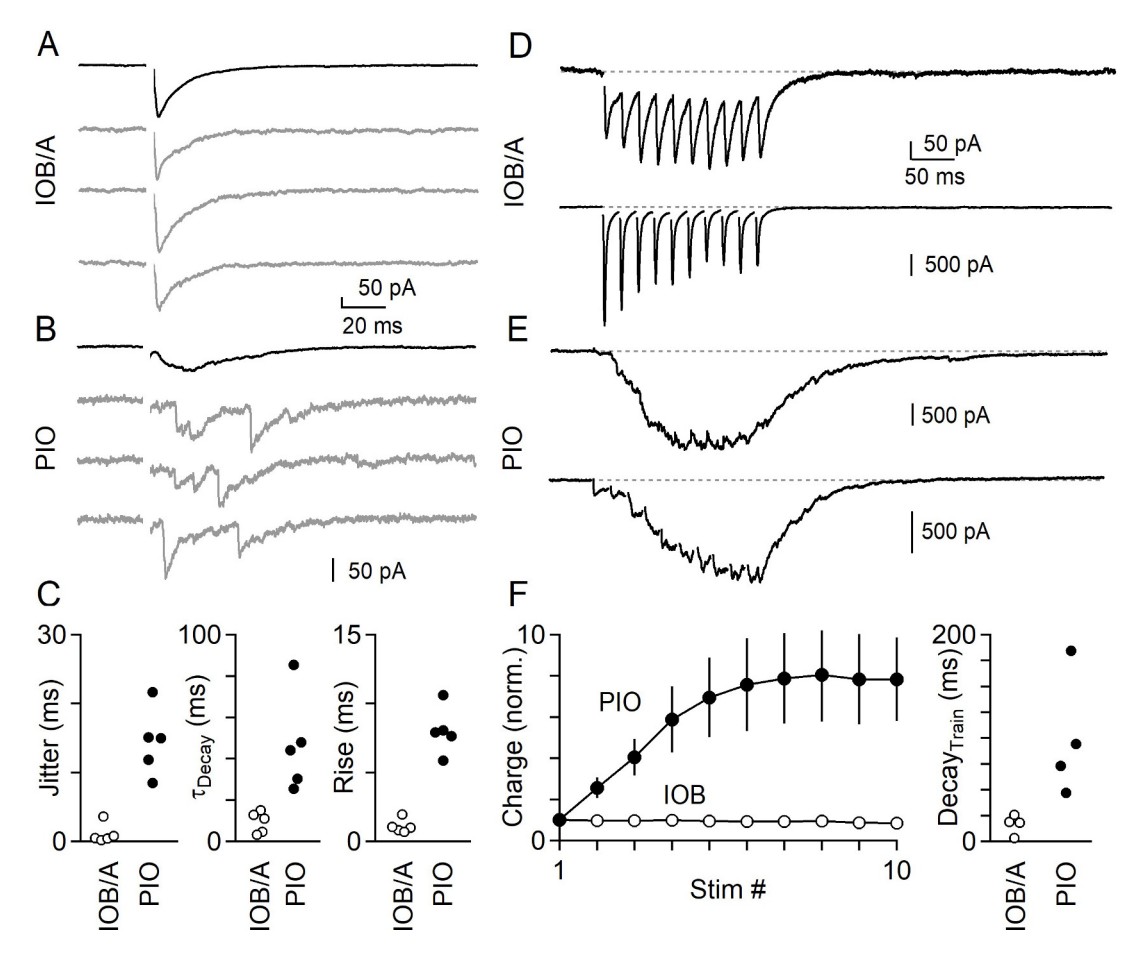

**Figure 6.** Synchronous and asynchronous release in the IO under near-physiological conditions (35°C in 1.5 mM $Ca^{2+}$/1.0 mM $Mg^{2+}$). (**A**) IPSCs evoked by single electrical stimuli in the IOB/A showing individual trials (gray) and average across trials (black). (**B**) Same as in A, but for the PIO. (**C**) Jitter (Left), decay time (middle) and rise time (right) of IPSCs in the IOB/A (empty markers) and PIO (filled markers) evoked by single stimuli in near-physiological conditions. (**D**) Averaged IPSCs evoked by trains of electrical stimuli (50 Hz x 10 stimuli) for two different IO neurons in the IOB/A. (**E**) Same as in D, but for two IO neurons in the PIO. (**F**) Average normalized charge transfer during a 50 Hz train (left) and average decay time of the last IPSC in the 50 Hz train (right) for neurons in the IOB/A (empty markers) and PIO (filled markers). Data is represented as mean ± SEM for charge, and individual cells for decay time.

The online version of this article includes the following source data and figure supplement(s) for figure 6:

**Source data 1.** Synchronous and asynchronous release in the IO under near-physiological conditions.

**Figure supplement 1.** bursting and oscillation of electrically-coupled neurons in the IO can obscure synaptic responses.

was little summation over the course of the train. The IPSC amplitude and total charge was either constant or underwent weak short-term depression, and release rapidly decayed after the last stimulus of the train (**Figure 6F**). In the PIO, inhibition evoked by high-frequency stimulation was very different. In the PIO asynchronous release became more prominent with each stimulus (**Figure 6E,F**, **Figure 6—source data 1**). Release was also delayed and did not decay between stimuli, resulting in a more tonic inhibition that after several stimuli generated a constant direct current. Following the train, release decayed slowly over the course of tens of milliseconds (**Figure 6F**).

The results suggest that inhibition originating from the DCN and VN are very different. Inhibition from the VN is rapid and phasic. During elevated activity, the strength of inhibition from the VN undergoes only minor use-dependent changes. Inhibition from the DCN is slow, does not decay between stimuli, and is much more prominent during trains of high-frequency activation.

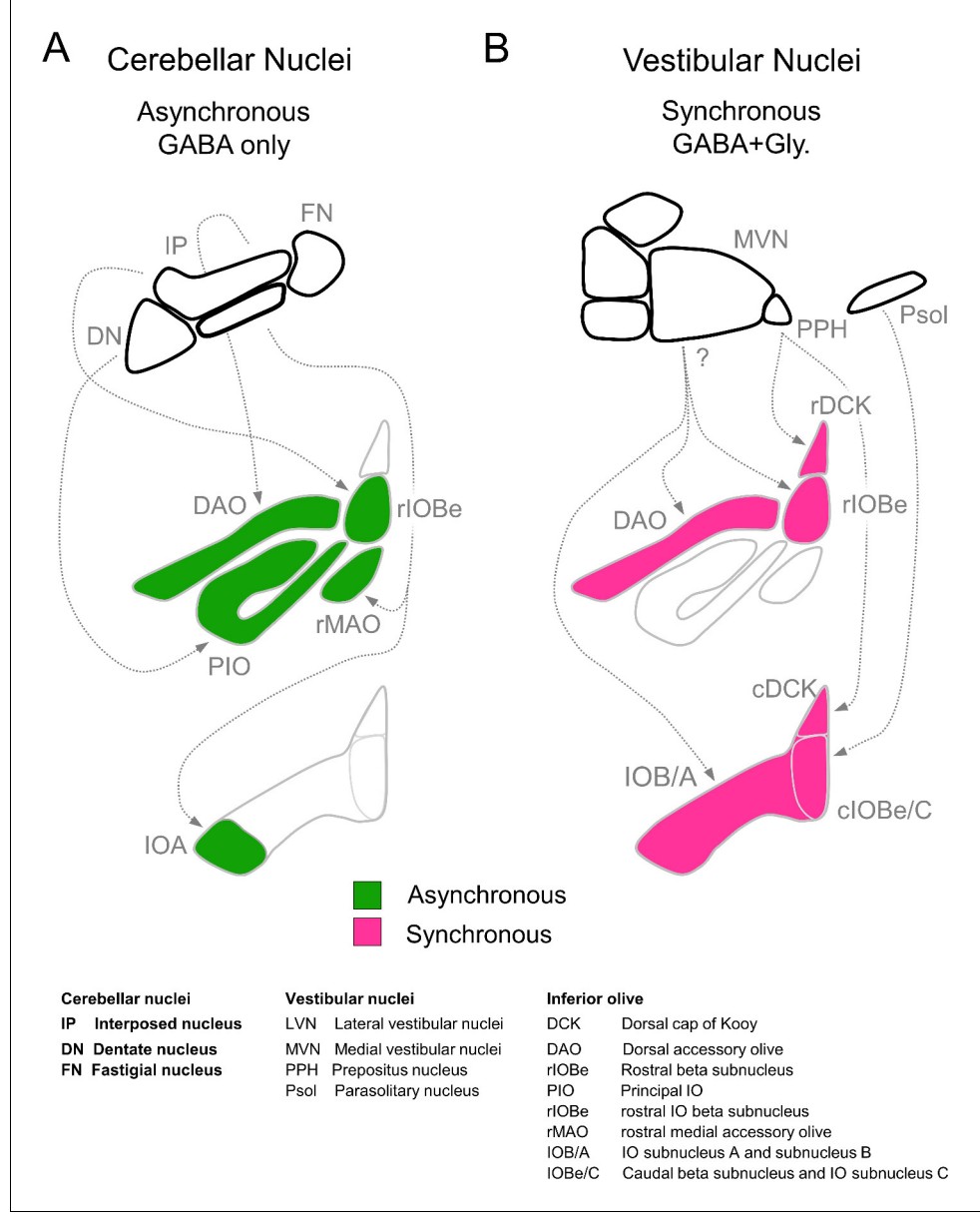

**Figure 7.** Summary of inhibitory projections to the IO. (**A**) Schematic of inhibitory input from the DCN (top) to the IO (middle, bottom). The DCN supplies asynchronous (green) GABAergic inhibition, primarily targeting the rostral IO. Dashed arrows show putative connections between specific subregions of the DCN and IO. White regions of the IO did not receive detectable input from the DCN. (**B**) Inhibitory input from vestibular origins (top) to the IO (middle, bottom) provide primarily synchronous (magenta) inhibition that typically has both GABAergic and Glycinergic components. White regions of the IO did not receive detectable input from the vestibular nuclei. Contralateral/ipsilateral specificity is not shown. Sources and properties of inhibition in the rDCK are based on previous studies, which have shown that they have fast kinetics (***Best and Regehr, 2009***), and originate in the vestibular nuclei but not the cerebellar nuclei (***Balaban and Beryozkin, 1994***; ***De Zeeuw et al., 1994***; ***De Zeeuw et al., 1993***).

## Discussion

We found that the two primary sources of inhibitory inputs to the IO are specialized to provide very different types of inhibition. DCN synapses are exclusively GABAergic, and lack Syt1/2, resulting in completely asynchronous release. VN synapses are mediated by both GABA and glycine, and express Syt1/2, leading to synchronized release.

## Determination of release kinetics in the IO

Our results provide insight into the determinants of the kinetics of transmitter release at inhibitory synapses in the IO. Prior to this study, a leading possible explanation for differences in the kinetics of inhibitory transmission across IO subnuclei was that subregions of the DCN differentially express Syt1/2. The observation that DCN inputs to the IO are all asynchronous, regardless of their targets in the IO, indicates that this is not the case. Another major issue was whether regions of the IO where electrically evoked inhibition had both synchronous and asynchronous components reflected the ability of individual boutons and individual axons to mediate both synchronous and asynchronous release. The properties of release from isolated DCN or VN inputs indicates that, in general, a given brain region provides primarily either synchronous or asynchronous release. Consequently, we can conclude that for inhibitory transmission in the IO, individual boutons and individual axons with multiple boutons generally do not mediate both synchronous and asynchronous release. The mixture of asynchronous and synchronous release evoked by electrical stimulation in some regions is a consequence of activating axons that originate from two different sources. In subnuclei such as the rIOBe, where asynchronous and synchronous inhibition are both prominent, electrical stimulation evokes inhibition consisting of two components: one from synchronous VN synapses, and the other from asynchronous DCN synapses.

Optogenetics studies and immunohistochemical analysis indicate that release kinetics are segregated by presynaptic origin. Asynchronous release at VN to IO synapses and synchronous release at DCN to IO synapses are exceedingly small, but it is difficult to completely exclude contributions from these components. A fraction of glycinergic boutons in the rDCK (20%) and the cIOBe/C (46%) appear to lack Syt1/2 (*Figure 1F*, *Figure 1—figure supplement 2*), which raises the possibility that some glycinergic synapses are asynchronous in these regions. Similarly, Syt1/2 appears to be present at a small fraction of exclusively GABAergic synapses in the cIOBe/C and the rIOBe (*Figure 1F*, *Figure 1—figure supplement 2*), suggesting that there is a synchronous component of release at a small fraction of exclusively GABAergic synapses in these regions. Interpreting these findings is complicated because regions other than the DCN and the VN provide some inhibitory inputs to the IO, and our anatomical experiments did not identify the presynaptic origin of inhibitory boutons. The contributions of glycine, GABA and Syt1/2 to the kinetics of release of inhibitory synapses from regions other than the DCN and VN are not known. It is therefore difficult to completely exclude the possibility that prominent synchronous release masks a tiny component of asynchronous release at VN to IO synapses, and similarly that prominent asynchronous release could mask a tiny component of synchronous release at DCN to IO synapses.

## Spatial extent of DCN and VN inputs to the IO

The optogenetic characterization of DCN and VN synapses combined with the immunohistochemical characterization of inhibitory synapses in the IO, allow us to estimate the regional distribution of DCN and VN inputs to the IO (*Figure 7*; *Anguat and Cicirata, 1982*; *Dietrichs and Walberg, 1986*; *Dietrichs et al., 1985*; *Pijpers et al., 2005*; *Ruigrok, 1997*; *Ruigrok and Voogd, 1990*). The finding that VN inputs are fast with a glycinergic component suggest that Syt1/2 and GlyT2 are present in these boutons (*Figure 1E*, GlyT2+, large $R^2$). In contrast, DCN inputs are asynchronous, exclusively GABAergic, and thus lack Syt1/2 and GlyT2 (*Figure 1E*, GlyT2−, small $R^2$). Thus, the GlyT2− boutons in *Figure 1C* provide a map of the DCN inputs to the IO. Exclusively GABAergic inputs lacking Syt1/2 are present throughout the rostral IO, but only account for a very small percentage of inputs to the rDCK. DCN inputs account for a large fraction of inputs in the rostral IO, and are present in subregions of the rIOBe and the cIOBe/C. VN inputs are prominent in all regions of the caudal IO, provide the bulk of the inputs to the cDCK, and provide a low percentage of inputs to some regions in the rostral IO.

Our optogenetic studies combined with our immunohistochemical studies suggest that VN makes much more extensive inhibitory projections to the IO than had previously been appreciated (*Figure 7B*). Retrograde and anterograde tracing techniques established that the VN projects to various IO subnuclei (*Barmack, 2006*; *Carleton and Carpenter, 1983*; *Gerrits et al., 1985*; *Saint-Cyr and Courville, 1979*; *Voogd and Barmack, 2006*), but in most cases the extent to which inhibitory neurons contributed to these projections had not been determined. It was however established that the parasolitary nucleus, which can be considered the most caudal region of the vestibular

complex, contains vestibular responsive inhibitory neurons that project to the IOBe and to the dorso-medial cell column of the IO (*Barmack, 1996*; *Barmack et al., 1993*; *Barmack et al., 1998*; *Nelson et al., 1989*). It has also been shown that the prepositus nucleus (PPH) provides vestibular inhibitory inputs to the DCK (*De Zeeuw et al., 1993*). Although our experiments do not identify the specific regions of the VN that project to the IO, our results confirm that the VN provide prominent inhibitory inputs to the DAO, the IOBe, and much of the caudal IO.

### GABA and glycine corelease

The differential kinetics of the glycinergic and GABAergic components, and the considerable spread in the contributions of these two components to VN to IO synapses, suggests that the regulation of these components could provide a means of adjusting the kinetics of inhibition within the IO. The relative strengths of glycinergic and GABAergic components is determined post-synaptically by the density of $GABA_A$ and glycine receptors, and presynaptically by vesicular content (*Apostolides and Trussell, 2013*; *Aubrey and Supplisson, 2018*; *Gamlin et al., 2018*; *Legendre, 2001*). Glycine and GABA share the transporter vesicular inhibitory amino acid transporter (VIAAT), also known as Vgat (*Aubrey, 2016*; *Aubrey et al., 2007*; *Aubrey and Supplisson, 2018*; *Wojcik et al., 2006*). VIAAT is nonspecific, and the relative concentrations of glycine and GABA in a vesicle is regulated by the cytosolic levels of GABA and glycine. Corelease of glycine and GABA also occurs at other synapses (*Dufour et al., 2010*; *Hirrlinger et al., 2019*; *Jonas et al., 1998*; *Keller et al., 2001*; *Medelin et al., 2016*; *Nabekura et al., 2004*; *Polter et al., 2018*; *Rahman et al., 2013*; *Rajalu et al., 2009*; *Russier et al., 2002*; *Wu et al., 2002*). For VN to IO synapses, and other synapses that corelease GABA and glycine, the contribution of GABA and glycine to inhibition could be regulated by the contents of vesicles or by the composition of postsynaptic glycine and GABA receptors.

### Functional consequences of specializations of DCN and VN synapses

Neurons in the inferior olive are electrically-coupled, and it is thought that one of the primary functions of inhibition in IO neurons is to regulate the extent of synchronous firing of IO neurons by regulating the effective gap junction coupling (*Best and Regehr, 2009*; *Lang et al., 1996*; *Lefler et al., 2014*). Some GAD-immunoreactive boutons that synapse onto IO dendrites are in close proximity to gap junctions, suggesting that they are well suited to this task (*de Zeeuw et al., 1988*; *de Zeeuw et al., 1989*; *de Zeeuw et al., 1990*; *Sotelo et al., 1986*). In the rMAO it was shown that many of the inhibitory boutons associated with gap junctions originated in the DCN (*de Zeeuw et al., 1989*), and our findings suggest that in mice the rMAO is inhibited exclusively by the DCN. Asynchronous release of GABA may be ideal to provide slow tonic inhibition that could shunt current away from gap junctions and effectively uncouple or desynchronize neurons in the IO. It has also been shown that some vestibular inhibitory inputs from the PPH to the DCK are present within glomeruli containing gap junctions (*De Zeeuw et al., 1993*), and may therefore also regulate electrical coupling between DCK neurons. It is unclear what fraction of synapses from each source target gap junctions, and how they ultimately influence the firing and coupling of IO neurons in different subnuclei. Since IO subnuclei are part of distinct cerebellar and vestibular circuits the kinetics of neurotransmitter release and inhibition may be suited to the needs of their unique functions.

## Materials and methods

**Key resources table**

| Reagent type (species) or resource | Designation | Source or reference | Identifiers | Additional information |
|---|---|---|---|---|
| Strain, strain background | C57BL/6, *Mus musculus* | Charles River | | |
| Strain, strain background *Mus musculus* | GlyT2$^{Cre}$ (C57BL/6-Slc6a5 < tm1.1(cre)Ksak>), C57BL/6, | Obtained from the RIKEN BioResource Research center; *Kakizaki et al., 2017* | BRC No: RBRC10109 | |
| Strain, strain background *Mus musculus* | Ai34 (B6;129S-*Gt(ROSA)26 Sor$^{tm34.1(CAG-Syp/tdTomato)Hze}$*/J) | Jackson Laboratories | Stock: 012570 | |

*Continued on next page*

*Continued*

| Reagent type (species) or resource | Designation | Source or reference | Identifiers | Additional information |
|---|---|---|---|---|
| Recombinant DNA reagent | AAV9-hSyn-hChR2 (H134R)-EYFP | UNC Viral Vector Core/Addgene | 26973-AAV9 | |
| Software, algorithm | Matlab | Mathworks (https://www.mathworks.com/downloads/) | RRID:SCR_001622 | Version R2019a |
| Software, algorithm | IgorPro | Wavemetrics (https://www.wavemetrics.com/order/order_igordownloads6.htm) | RRID:SCR_000325 | Version 6.22 |
| Software, algorithm | ImageJ/Fiji software | Fiji (http://fiji.sc/) | RRID:SCR_002285 | Version 1.52N |
| Software, algorithm | ZEN software | Zeiss (https://www.zeiss.com/microscopy/us/products/microscope-software/zen.html) | RRID:SCR_013672 | Version 2.3 SP1 FP3 |
| Chemical compound, drug | Tetrodotoxin citrate | Abcam | Ab120055 | |
| Chemical compound, drug | NBQX disodium salt | Abcam | Ab120046 | |
| Chemical compound, drug | (R)-CPP | Abcam | Ab120159 | |
| Chemical compound, drug | Strychnine hydrochloride | Abcam | Ab120416 | |
| Chemical compound, drug | SR95531 (Gabazine) | Abcam | Ab120042 | |
| Chemical compound, drug | TTA-P2 | Alamone Labs | T-155 | |

## Animals

Adult ($\geq$P40) animals of both sexes were used for experiments. For electrophysiology, C57/BL6 were obtained from Charles River. For anatomical experiments, GlyT2$^{Cre}$ animals were obtained from the RIKEN BioResource Research Center and crossed with R26$^{LSL-Synaptophysin-tdTomato}$ (Ai34, Jackson labs) mice. Experiments were not performed blind to viral injections. The number of experiments and the number of animals used for physiology experiments are listed in *Table 1*.

## Slice preparation

Slices were made from adult ($\geq$P40) C57BL/6 mice obtained from Charles River. Animals were anesthetized with isoflurane and transcardially perfused with solution composed of in mM: 110 Choline Cl, 2.5 KCl, 1.25 $NaH_2PO_4$, 25 $NaHCO_3$, 25 glucose, 0.5 $CaCl_2$, 7 $MgCl_2$, 3.1 Na Pyruvate, 11.6 Na Ascorbate, 0.002 (R,S)-CPP, 0.005 NBQX, oxygenated with 95% $O_2$/5% $CO_2$, kept at 35℃. Vertebrae and the back of the skull was dissected away and the hind brain was isolated by making a cut between the border of the cerebellum and midbrain. The dura was then carefully removed from the brainstem and the cut face was glued down. 200 µm thick coronal sections of the brainstem were made using a Leica 1200S vibratome and then transferred to a chamber with ACSF containing in mM: 127 NaCl, 2.5 KCl, 1.25 $NaH_2PO_4$, 25 $NaHCO_3$, 25 glucose, 1.5 $CaCl_2$, 1 $MgCl_2$, allowed to recover at 35℃ for at least 20 min, and then stored at room temperature in the same solution. For experiments involving AAV, experiments were performed 14–28 days after injections. Coronal slices of the entire hindbrain were collected. Following electrophysiology experiments slices were fixed in 4% PFA in PBS for 2 hr at room temperature. Images of sections containing the DCN and VN were collected on a Zeiss Axioscope under a 5x objective to confirm restriction of the injection to either the DCN or dorsal brainstem for all experiments.

## Electrophysiology

Experiments were performed at room temperature using a flow rate of 3–5 ml/min. In order to avoid triggering subthreshold oscillations and bursting in electrically-coupled IO neurons, single stimuli were used, and the ACSF used for recording had the same composition as incubation ACSF except

**Table 1.** Number of electrophysiological experiments.

| Figure | Sub nucleus | Electrical/No stim. (74 Cells/21 Mice) | ChR2; VN (54 Cells/11 Mice) | ChR2; DCN (51 Cells/9 Mice) | Drugs (+NBQX, CPP) |
|---|---|---|---|---|---|
| *Figure 2A-C* | DAO | 6/2 | | | Gbz wash, Strych wash |
| *Figure 2D-H* | DAO | 15/5 | | | TTX, Gbz |
| *Figure 2D-H* | rIOBe | 6/3 | | | TTX, Gbz |
| *Figure 2D-H* | rMAO | 4/2 | | | TTX, Gbz |
| *Figure 2D-H* | PIO | 12/5 | | | TTX, Gbz |
| *Figure 2D-H* | cDCK | 4/2 | | | TTX, Gbz |
| *Figure 2D-H* | cIOBe/C | 6/3 | | | TTX, Gbz |
| *Figure 2D-H* | IOB/A | 6/3 | | | TTX, Gbz |
| *Figure 2D-G* | DAO | *Turecek and Regehr, 2019* | | | TTX, Strych |
| *Figure 3B* | rIOBe | 1/1 | | | None |
| *Figure 4* | DAO | | 13/5 | 12/4 | None |
| *Figure 3C-F; Figure 4* | rIOBe | | 7/7 | 9/3 | None |
| *Figure 4* | rMAO | | 3/3 | | None |
| *Figure 4* | PIO | | 12/5 | | None |
| *Figure 4* | cDCK | | 0/0 | | None |
| *Figure 4* | cIOBe/C | | 9/2 | 8/3 | None |
| *Figure 4* | IOB/A | | 5/2 | 17/6 | None |
| *Figure 5A-C* | PIO | | 5/2 | | Gbz wash, Strych wash |
| *Figure 5D-E* | IOB/A | | | 5/2 | Gbz wash, Strych wash |
| *Figure 6* | PIO | 4/4 | | | None |
| *Figure 6* | IOB/A | 4/4 | | | None |
| *Figure 6* | PIO | 5/5 | | | None |
| *Figure 6* | IOB/A | 5/5 | | | None |

$Ca^{2+}$ was raised to 2.5 mM and $Mg^{2+}$ lowered to 0.5 mM to maximize evoked release by single stimuli. All experiments were performed in 5 µM NBQX to block AMPARs, and 2.5 µM (R)-CPP to block NMDARs. Borosilicate electrodes (1–2 MΩ) contained internal solution consisting in mM of: 110 CsCl, 10 HEPES, 10 TEA-Cl, 1 $MgCl_2$, 4 $CaCl_2$, 5 EGTA, 20 Cs-BAPTA, 2 QX314, 0.2 D600, pH to 7.3. Cells were held at −60 mV. Conditions were adjusted to minimize subthreshold membrane potential oscillations (*Turecek and Regehr, 2019*; *Best and Regehr, 2009*; *Figure 6—figure supplement 1*) that are generated by IO neurons (*Llinás and Yarom, 1981a*; *Llinás and Yarom, 1981b*; *Llinás and Yarom, 1986*; *Turecek et al., 2016*). 1 µM TTA-P2 was included in the ACSF to block t-type $Ca^{2+}$ channels, and experiments were performed at room temperature to suppress oscillations. For experiments requiring electrical stimulation, a glass monopolar electrode (2–3 MΩ) filled with ACSF was placed several hundred µm away within the IO or in the surrounding reticular formation. Neurons in the IO do not form chemical synapses within the IO, and the overwhelming majority of IO neurons do not express markers for the synthesis or release of GABA.

Axons expressing ChR2 were stimulated by 473 nm light from an LED (Thorlabs) that was guided through a 60x objective producing an 80 µm diameter spot positioned several hundred µm away from the recorded cell. Single 0.5–1 ms pulses were used with an intensity of 80 mW/mm² measured under the objective. All data are presented as individual cells unless otherwise noted. Whole-cell capacitance and series resistance were uncompensated for all experiments. All currents were reversible at 0 mV, indicating that they were not mediated through gap junctions. Measurements of mIPSC frequency were performed in the presence of TTX (0.5 µM) and in slices unperturbed by stimulus electrodes. Experiments measuring GABA and glycinergic components of release were performed in the presence of 5 µM SR95531 (Gabazine) to isolate glycinergic release, or 0.5 µM strychnine to

isolate GABA release. For *Figure 2F–G* mIPSCs measured in the presence of gabazine were compared to mIPSCs measured in strychnine from previously published work (*Turecek and Regehr, 2019*).

A subset of experiments (*Figure 6*) were performed under near -physiological conditions. For these experiments, the ACSF composition was the same as above except divalents were adjusted to (in mM): 1.5 CaCl$_2$, 1 MgCl$_2$. The bath solution was warmed using an in-line heater to 35˚C. Bursting and subthreshold oscillations are much more prominent in the IO at warm temperatures, and they often contaminated recordings (*Figure 6—figure supplement 1*). Contamination by gap-junctional currents were minimized using several strategies. Cells were selected for in which spontaneous bursting was infrequent. Electrically-coupled bursts could be identified by their distinct waveform, and trials in which they were detected were discarded. For trains of stimuli, in some cases small consistent gap-junctional bursts were measured at the reversal potential for Cl- (0 mV) and were averaged, scaled and subtracted from the averaged trials for the same cell held at −60 mV. In some cases, Gabazine and strychnine were washed in and the remaining currents in the presence of blockers was subtracted.

## Viruses

AAV9-hSyn-hChR2(H134R)-EYFP viral preparations were obtained from the University of Pennsylvania Vector Core, or Addgene.

Stereotaxic surgeries were performed on adult (≥P40) C57/BL6 male mice anesthetized with ketamine/xylazine (100/10 mg/kg) supplemented with isoflurane (1–4%). Viruses were injected through fine-tipped glass capillary needles via a Nanoject III (Drummond) mounted on a stereotaxic (Kopf). Bilateral injections were made using stereotactic coordinates targeting various nuclei of the DCN, 1.5–2.5 mm lateral, 2 mm posterior from lambda, 2.5 mm depth. VN injections were performed by inserting the injection pipette at an angle to avoid passage through the cerebellum. An incision was made in the skin of the dorsal neck and neck muscle were cut or moved away from the midline to expose the dorsal surface of the brainstem. A small incision was made in the dura between the cerebellum and brainstem. The injector pipette mounted on the stereotactic frame was angled to pass along just below the dorsal surface of the brainstem and the tip was inserted at a depth of 2 mm. For DCN and VN injections 200–300 nL of virus suspension (2–5 × 10$^{13}$ vg/mL) was delivered to each site at a rate of 100 nL/minute, and the needle was retracted 10 min following injection. Subcutaneous analgesic (buprenorphine 0.05 mg/kg) was administered for 48 hr post-surgery. Injections were targeted to the DCN, but YFP was often detected in the cerebellar cortex just superficial to the DCN. Animals receiving injections in the DCN in which YFP was detected in the dorsal VN were discarded. Injections of the VN often labeled structures in the dorsal brainstem immediately surrounding the VN, and experiments in which expression of YFP was detected in DCN were discarded.

## Immunohistochemistry

One adult (≥P40) male GlyT2$^{Cre}$; R26$^{LSL-Synaptophysin-tdTomato}$ (Ai34) and one wildtype C57/BL6 male injected with AAV9-hSyn-ChR2-YFP were anesthetized with isoflurane and perfused transcardially with PBS, then with 4% paraformaldehyde (PFA) in PBS. The hindbrain was removed and post-fixed overnight in 4% PFA. The hindbrain was mounted in 6% low-melting agarose and coronal slices were made on a Leica VT1000S vibratome (30–50 μm). For immunohistochemistry, two sections were selected to capture the rostral and caudal IO using the same criteria used for physiology experiments (see Quantification and Statistical Analysis). Free-floating slices were permeabilized (0.2% triton X-100 in PBS) for 10 min followed by blocking for 1 hr (4% Normal Goat Serum in 0.1% triton X-100) at room temperature. Sections were then incubated overnight at 4˚C with primary antibodies (Mouse anti-Syt1, Synaptic Systems 105011, 1:500; Mouse anti-Syt2, Zirc znp-1, 1:200; Guinea-pig anti-VGAT, Synaptic Systems 131004, 1 μg/mL, 1:200). For all experiments, antibodies for Syt1 and Syt2 were applied together and labeled with the same secondary. Slices were then incubated with secondary antibodies for 2 hr at room temperature (anti-Guinea-pig-AlexaFluor488, Abcam ab150185; anti-Mouse-AlexaFluor647, Abcam ab150115). Tissue was mounted with #1.5 precision coverslips and Prolong Diamond Antifade mounting medium. Z-Stacks were collected with a Zeiss LSM-710 confocal microscope equipped with ZEN software. Images were collected using a 60 × 1.4 NA oil immersion objective. Mosaics of the ventral brainstem were collected and stitched using

ImageJ. Images were collected at a resolution of 0.1 µm/pixel with eight bit depth and z-spacing of 0.2 µm/section. Individual stacks were 101.6 × 101.6×3 µm in size. Stacks were collected within the top 5 µm of tissue where antibody penetration was most effective.

## Quantification and statistical analysis

### Electrophysiology analysis

Recordings were collected using a Multiclamp 700B (Molecular Devices) with Igor Pro software (Wavemetrics). Data was sampled at 20 kHz and filtered at 4 kHz. Analysis was performed using custom-written scripts in Matlab (Mathworks). All electrical stimulus artifacts were blanked for clarity. The average synaptic current was used to measure release kinetics including the half-rise time, decay time constant. Jitter was calculated by measuring the time of peak current amplitude for each individual trial and then calculating the standard deviation of peak times. mIPSC events were detected using a first derivative and integration threshold using custom-written scripts in Matlab. For all recordings, an image of the pipette tip position in the IO were taken under a 5x objective by a CCD. Images were then used to assign cells to a common map of the IO (*Fu and Watson, 2012*; *Paxinos and Franklin, 2001*; *Yu et al., 2014*). Although the IO is a 3D structure, a rostral and caudal coronal section of the IO were selected in order to analyze the largest number of regions that could be reliably identified in acute slices. Acute slices and fixed tissue were selected to match these common sections of the IO. Caudal sections of the IO were required to have lateral arms of the IO visible as single bands of gray matter and no parts of the DAO or PIO. Rostral slices were selected in which the IOBe and DCK were visible and the lateral aspect of the PIO and DAO were clearly defined. All physiology data are presented as individual cells unless otherwise noted.

### Anatomical and co-localization analysis

Analysis of images were performed as previously described (*Turecek and Regehr, 2019*). Images were not de-noised or processed after acquisition. Raw image data was collected as tiles that were individually analyzed using custom-written scripts in Matlab. Mosaic images for display were prepared in ImageJ using the Grid/Collection Stitching plugin (*Preibisch et al., 2009*). In some cases global changes to brightness were made for clarity.

Vgat-positive boutons were first identified by determining local maxima in each imaged 3D volume. For each identified synapse, a 1.5 × 1.5×1 µm volume of interest was isolated around the bouton centroid. Vgat and Syt1/2 signals were collected for each voxel and plotted against each other. Voxels on the edge of the region of interest (outer 0.25 µm) with Vgat signals generated by neighboring boutons were detected by a set intensity threshold and ignored. Vgat and Syt1/2 signals were correlated using linear regression, generating a value of $R^2$ as a measure of co-localization. For anti-correlated signals (slope of regression line <0) $R^2$ was set to 0.01.

Vgat-positive boutons were subdivided by whether they co-expressed synaptophysin-TdTomato (Syp-TdT) to isolate synapses that were glycinergic (Syp-TdT−positive, GlyT2+) or exclusively GABAergic (Syp-TdT−negative, GlyT2-). The presence of Syp-TdT was assessed using Vgat and Syp-TdT signal correlation performed identically as for Vgat and Syt1/2. After obtaining $R^2$ values for Vgat-Syp-TdT, a set threshold was used to isolate GlyT2+ boutons ($R^2_{Vgat-Syp-TdT} \geq 0.4$) and GlyT2- boutons ($R^2_{Vgat-Syp-TdT} \leq 0.1$).

Experiments examining YFP-labeled boutons originating from the DCN were performed similarly. Vgat-positive boutons in the DAO were identified as described above. The presence of YFP was assessed using the signal correlation between Vgat and YFP identically as for Vgat and Syt1/2. A set threshold was used to isolate YFP+ boutons ($R^2_{Vgat-YFP} \geq 0.4$). These boutons were then isolated analyzed for their co-localization with Syt1/2 as described above. To determine whether Syt1/2-containing Vgat boutons expressed YFP, we selected Vgat boutons that co-localized with Syt1/2 ($R^2_{Vgat-Syt1/2} \geq 0.4$) and then analyzed those boutons for their co-localization with YFP using the same correlation approach described above ($R^2_{Vgat-YFP}$).

Subnuclei of the IO were outlined and defined by visual inspection of mosaic images. A 170 × 170×3 µm volume just dorsal to the DAO was sampled as the reticular formation. Some nuclei were grouped because they could not be reliably distinguished, and could not be compared across experiments, especially in acute brainstem slices. The ventrolateral outgrowth could not be reliably distinguished from the rostral beta subnucleus. For others, borders could not be clearly defined in

acute slices such as for caudal beta subnucleus, subnucleus C and more recently defined subnucleus D (*Fujita et al., 2020*), or for the caudal subnuclei A and B, and were therefore grouped for analysis. The dorsal fold of the DAO was not examined.

### Data and software availability

Source data files for all figures are provided, and custom-written Igor and Matlab analysis scripts will be made available at github, upon acceptance of the manuscript (https://github.com/josefturecek/Elife_IO_inhibitory_inputs).

# Acknowledgements

We thank the Regehr lab for comments on the manuscript. This work was supported by National Institutes of Health Grant R35NS097284 to WGR.

# Additional information

### Funding

| Funder | Grant reference number | Author |
| --- | --- | --- |
| NIH Office of the Director | R35NS097284 | Wade G Regehr |

The funders had no role in study design, data collection and interpretation, or the decision to submit the work for publication.

### Author contributions

Josef Turecek, Conceptualization, Data curation, Formal analysis, Investigation, Methodology, Writing - original draft; Wade G Regehr, Conceptualization, Resources, Supervision, Funding acquisition, Writing - original draft

### Author ORCIDs

Josef Turecek (iD) https://orcid.org/0000-0001-9843-1877
Wade G Regehr (iD) https://orcid.org/0000-0002-3485-8094

### Ethics

Animal experimentation: All animal procedures were carried out in accordance with the NIH and Animal Care and Use Committee (IACUC) guidelines and protocols approved by the Harvard Medical Area Standing Committee on Animals (animal protocol #1493).

### Decision letter and Author response

Decision letter https://doi.org/10.7554/eLife.61672.sa1
Author response https://doi.org/10.7554/eLife.61672.sa2

# Additional files

### Supplementary files

• Transparent reporting form

### Data availability

All data generated or analysed during this study are included in the manuscript and supporting files. Source data files have been provided.

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
