## [Decision Letter]

**Acceptance summary:**

Previous work has demonstrated a diversity of kinetics at inhibitory synapses within the inferior olive, a brain region that sends climbing fiber projections to cerebellar Purkinje cells. This paper demonstrates that two distinct inputs to the inferior olive, the deep cerebellar and vestibular nuclei, provide inhibitory inputs with distinct kinetics, a finding that could have important implications for signaling within the cerebello-olivary network. In addition, this work provides further evidence that synaptotagmin isoforms determine the mode of synaptic vesicle release.

**Decision letter after peer review:**

Thank you for submitting your article "Cerebellar and vestibular nuclei synapses within the inferior olive have distinct release kinetics and neurotransmitters" for consideration by *eLife*. Your article has been reviewed by three peer reviewers, and the evaluation has been overseen by a Reviewing Editor and Richard Aldrich as the Senior Editor. The following individual involved in review of your submission has agreed to reveal their identity: Jaideep Singh Bains (Reviewer #1).

The reviewers have discussed the reviews with one another and the Reviewing Editor has drafted this decision to help you prepare a revised submission.

This work is an extension of the 2019 Turecek and Regehr paper in Neuron that explored synchronous and asynchronous inhibitory transmission in the inferior olive. Here the focus was on contrasting distinct sources of GABA vs. glycine and whether the kinetics of transmission at those synapses differ. A major conclusion is that input from the cerebellar nuclei is GABAergic and releases vesicles asynchronously, while input from vestibular nuclei releases synchronously and uses dual GABA/glycine transmission. The mapping of these inputs through the different IO subnuclei is determined, and the overlap with synaptotagmin isoforms 1/2 is also mapped out. These findings will be important on at least two fronts, 1) this work provides further evidence that synaptotagmin 1/2 expression determines the mode of vesicle release, and 2) the remarkable differences in synaptic kinetics will be important for understanding signaling and control in the cerebello-olivary network.

There was consensus among the reviewers that the work was well done and potentially interesting. However, the experimental conditions used for the recordings were atypical and there was concern about how the non-physiological temperature and high calcium would influence the results and interpretation of the data. It was recognized that these choices were likely related to the difficulties of recording from the inferior olive, and to minimize oscillations. There was also substantial discussion during the consultation period about whether recording at room temperature would over- or underestimate the differences in release mode between the two synapses. However, given these concerns, there was consensus that these issues need to be addressed and statements of physiological relevance in vivo should be tempered.

Essential revisions:

1) Please provide a clearer rationale for the experimental conditions and provide additional information in the discussion about the limitations and caveats in the study. Specifically, the limitations of recording at room temperature and possible effects on synchronous vs. asynchronous release should be discussed. In addition, if the authors happen to have data in lower Ca or higher temp, then that could be added. Points a) and b) below are the original comments of reviewer 2, which should be addressed.

a) A major concern is that the conclusions regarding release and IPSC kinetics are so dependent on the unusual experimental conditions as to make the work very difficult to generalize to the in vivo setting and to the functional significance of these synaptic differences. Three conditions in particular are of concern: 1) All recordings were at room temperature, 2) the bath contained 2.5 mM Ca^2+^ (twice normal levels), 3) voltage clamp did not employ series resistance compensation.

Recordings at room temperature will of course slow down the IPSCs, as will lack of Rs compensation. The physiological decay kinetics and how they differ remains unknown. Of even greater concern is the well-described effect of low temperature on facilitating asynchronous release. As shown by Huson et al., 2019 (and others), elevating recording temperature to the physiological level profoundly attenuates asynchronous release. The authors must know this work, given the recent Huson and Regehr review in Curr. Opin Neurobiol. Presumably this effect of temperature relates to Ca^2+^ handling in the terminal (or perhaps also tagmin kinetics). Elevating bath Ca^2+^ to 2.5 mM could only exacerbate the degree of asynchronous release, as it would prolong clearance from the terminals.

Accordingly, while the distinctions between GABA and glycine release outlined here are very clear, the conclusions about differences in kinetics among the different inputs are artificially magnified. (Editor note: Here were some differences of opinion among the reviewers, with reviewer 3 stating, "Regarding the Huson paper, my reading is that asynchronous release is increased with raising temperature at synapses lacking syt1, but decreased at synapses expressing syt1. This makes me think that DCN inputs to IO (presumably lacking syt1) would show greater asynchronous release at physiological temperature, and vestibular inputs (presumably containing syt1) would show less asynchronous release. If this is the case, recording at room temperature may underestimate the differences in release mode between these two synapses." The authors should address these issues directly.)

b) It is remarkable that such dramatic asynchronous release could be obtained with single stimuli rather than trains of shocks. It seems likely the experimental conditions enabled this to happen, but that under more normal ionic and temperature conditions and with trains, release might be relatively synchronous at first then decay into an later asynchronous mode.

Additional concerns:

1) More explanation for interpretation of R^2^ values for colocalization. For example how is an R^2^ value of 0.5 to be interpreted? The illustrated boutons in 1C show discontinuous labeling across the structure yet the R^2^ versions below show continuous value. Does 0.5 mean only half the bouton showed colocalization? Also, how does variation in antibody penetrance factor in to this variation? In addition, the authors did not clarify whether the results in Figure 1E were obtained from one animal or multiple animals. How many animals were used in this experiment?

2) Authors interpret the slow IPSCs purely as a presynaptic phenomenon, i.e., asynchronous release, without considering the potential contribution of pooling and spillover, which has been shown for GABA and glycine synapses in the past.

3) In Figure 6B, rDCK was labeled in red which means it receives synchronous inputs. However, no electrophysiological data support this idea (see Figure 4D). The authors need to add more data or remove the red color from rDCK.

4) The conclusion in the first paragraph of the Discussion: "DCN synapses are exclusively GABAergic, and lack Syt1/2, resulting in completely asynchronous release. VN synapses are mediated by both GABA and glycine, and express Syt1/2, leading to synchronized release." However, the authors showed no direct evidence between the expression of Syt1/2 and the source of inputs; all evidences are circumstantial. To make this conclusion more valid, it is necessary to check if the expression of EYFP (as a result of viral injection to DCN or VN) co-localizes with Syt1/2.

5) In the Materials and methods, both AAV2 and AAV9 were used, which is not consistent with the information listed in the Key Resources Table. In addition, the source of the virus is inconsistent between the table and the text.

6) Please rewrite the following sentence in the Materials and methods: "Jitter was calculated by measuring the time of peak current amplitude for each individual trial was measured and calculating the standard deviation of peak times."

7) Much of this work depends on the accuracy of viral expression of ChR2 in the vestibular or deep cerebellar nuclei. More quantification of YFP expression in the target regions and surrounding areas would be helpful. In Figure 3C it appears there is also high YFP expression in the cerebellar cortex, was this generally true?

8) In previous work the same group has shown that synaptic responses from optogenetic stimulation of presynaptic fibers can differ from responses evoked by electrical stimulation, depending on cell-type. This raises the possibility that optogenetic stimulation affects responses from DCN terminals differently than responses from vestibular terminals, possibly accounting for the observed differences in synaptic properties, at least in part. This potential issue should be discussed briefly.

9) I find this study well executed. I have no concerns about the dataset, analysis or interpretation. It is intriguing that 2 different inputs use different strategies to manipulate the temporal window of inhibition. Conceptually, I find this idea fascinating and perhaps the authors could consider a short Discussion point on this.

10) Do the authors have any information on whether individual vesicles released at VN terminal onto IO cells contain both GABA and glycine? One possibility would be to assess the decay of mIPSCs. In some cases, there would be a fast monoexponential decay (glycine), in others, a slower monoexponential (GABA) and in a third population, perhaps a double exponential decay (glycine and GABA). This is more a curiosity on my part and not a requirement for the study.

11) The Huson paper uses trains of stimuli and, here, I think things are very different. The authors could discuss this issue (single stim vs. multiple stim) and consequences for synapses that release synchronously vs. asynchronously.

12) The title reads awkwardly with 'nuclei' modifying synapses, almost as if the nuclei themselves are within the inferior olive. Suggest using the more grammatically correct 'nuclear' instead, or rearranging the components of the title to clarify that the synapses are from those nuclei.

---

## [Author Response]

[…] There was consensus among the reviewers that the work was well done and potentially interesting. However, the experimental conditions used for the recordings were atypical and there was concern about how the non-physiological temperature and high calcium would influence the results and interpretation of the data. It was recognized that these choices were likely related to the difficulties of recording from the inferior olive, and to minimize oscillations. There was also substantial discussion during the consultation period about whether recording at room temperature would over- or underestimate the differences in release mode between the two synapses. However, given these concerns, there was consensus that these issues need to be addressed and statements of physiological relevance in vivo should be tempered.

We thank Dr. Carey, Dr. Bains and the other reviewers for their helpful comments. We agree with the reviewers that in general it is important to study synapses under physiological conditions. In our original studies of inhibitory synapses in the IO (which were done in rat) we performed experiments at near physiological temperatures (Best and Regehr, 2009). Those studies were very challenging because of the strong electrical coupling between IO neurons, which allows IO neurons to generate subthreshold oscillations and bursts that can interfere with studies of synaptic responses.

In our recent study (Turecek and Regehr, 2019) we moved to mice to determine the mechanisms that control the kinetics of transmitter release, and we felt that the best way to characterize the kinetics of neurotransmitter release was to study synaptic currents evoked by single stimuli. Trains of electrical stimuli can generate bursting and oscillations from electrically-coupled olivary neurons that would contaminate recordings, so we maximized asynchronous release from single stimuli by performing experiments in 2.5 mM Ca. We also preferred single stimuli for optogenetic experiments, where it is sometimes difficult to reliably activate presynaptic fibers with high-frequency trains. Subthreshold oscillations and bursting from electrically-coupled olivary neurons are more prominent at physiological temperatures. Even if oscillations are absent at room temperature, oscillations will often emerge when slices are warmed to physiological temperatures.

We now more extensively describe the rationale for our experimental approach. We also discuss the issue of oscillations and provide a supplementary figure highlighting this problem (Figure 6—figure supplement 1). We performed additional experiments in physiological calcium at elevated temperatures to address these concerns (Figure 6). As a result of oscillations most synaptic experiments performed in physiological conditions had to be discarded. Electrical stimulation was employed because we were concerned about using optogenetic approaches to stimulate with trains, as we discussed here and illustrated in previous work (Jackman et al., 2014). The additional experiments were performed using electrical stimulation in regions where either fast or slow inhibitory inputs predominate.

Essential revisions:1) Please provide a clearer rationale for the experimental conditions and provide additional information in the discussion about the limitations and caveats in the study. Specifically, the limitations of recording at room temperature and possible effects on synchronous vs. asynchronous release should be discussed. In addition, if the authors happen to have data in lower Ca or higher temp, then that could be added. Points a) and b) below are the original comments of reviewer 2, which should be addressed.a) A major concern is that the conclusions regarding release and IPSC kinetics are so dependent on the unusual experimental conditions as to make the work very difficult to generalize to the in vivo setting and to the functional significance of these synaptic differences. Three conditions in particular are of concern: 1) All recordings were at room temperature, 2) the bath contained 2.5 mM Ca^2+^ (twice normal levels), 3) voltage clamp did not employ series resistance compensation.Recordings at room temperature will of course slow down the IPSCs, as will lack of Rs compensation. The physiological decay kinetics and how they differ remains unknown. Of even greater concern is the well-described effect of low temperature on facilitating asynchronous release. As shown by Huson et al., 2019 (and others), elevating recording temperature to the physiological level profoundly attenuates asynchronous release. The authors must know this work, given the recent Huson and Regehr review in Curr. Opin Neurobiol. Presumably this effect of temperature relates to Ca^2+^ handling in the terminal (or perhaps also tagmin kinetics). Elevating bath Ca^2+^ to 2.5 mM could only exacerbate the degree of asynchronous release, as it would prolong clearance from the terminals.Accordingly, while the distinctions between GABA and glycine release outlined here are very clear, the conclusions about differences in kinetics among the different inputs are artificially magnified. (Editor note: Here were some differences of opinion among the reviewers, with reviewer 3 stating, "Regarding the Huson paper, my reading is that asynchronous release is increased with raising temperature at synapses lacking syt1, but decreased at synapses expressing syt1. This makes me think that DCN inputs to IO (presumably lacking syt1) would show greater asynchronous release at physiological temperature, and vestibular inputs (presumably containing syt1) would show less asynchronous release. If this is the case, recording at room temperature may underestimate the differences in release mode between these two synapses." The authors should address these issues directly.)

Most of these issues are dealt with in our general response and our new findings. With regard to the specific paper by Huson et al., they found that in cultured cells asynchronous release evoked by trains of activity became much more prominent at low temperature. They did not see such an effect for single stimuli. We have added new figures to explain our rationale for performing most experiments at room temperature with single stimuli (Figure 6—figure supplement 1), and we have performed additional experiments in physiological calcium at near physiological temperatures (Figure 6).

b) It is remarkable that such dramatic asynchronous release could be obtained with single stimuli rather than trains of shocks. It seems likely the experimental conditions enabled this to happen, but that under more normal ionic and temperature conditions and with trains, release might be relatively synchronous at first then decay into an later asynchronous mode.

We find that release can be entirely asynchronous. This appeared to be the case in previous work performed at near-physiological temperature in rats (Best and Regehr, 2009), and also appears to be true in our new experiments performed in physiological calcium at elevated temperatures in mice (Figure 6).

Additional concerns:1) More explanation for interpretation of R^2^ values for colocalization. For example how is an R^2^ value of 0.5 to be interpreted? The illustrated boutons in 1C show discontinuous labeling across the structure yet the R^2^ versions below show continuous value. Does 0.5 mean only half the bouton showed colocalization? Also, how does variation in antibody penetrance factor in to this variation? In addition, the authors did not clarify whether the results in Figure 1E were obtained from one animal or multiple animals. How many animals were used in this experiment?

The top 4 panels of Figure 1C show raw images of immunostaining, whereas the analyzed images below are identified boutons that are color coded for the correlation coefficient between Vgat and Syt signal for the voxels of the bouton. R^2^ is a measure of how correlated Vgat and Syt are for a given bouton. The analysis is performed for each bouton. A small volume around each bouton is taken, and the intensity of Syt fluorescence is plotted vs. Vgat intensity for all voxels. A linear fit is made to the resulting plot, and the correlation coefficient R^2^ is used as a measure of co-localization. We illustrated the co-localization analysis in our previous paper (Turecek and Regehr, 2019, Supplementary Figure 3), where we provided a detailed description of the method. We provide a basic description of the methods in the paper and provide a reference to the detailed methods in our previous paper.

We do not think that tissue penetration is an issue. We collected anatomical data from the top 5 µm of tissue where antibodies abundantly penetrate the tissue. We now point out that two animals were used for fluorescence image analysis, one animal for Figure 1 and a second for Figure 4—figure supplement 1. The distribution of Vgat-Syt1/2 co-localization in the IO matched that of two other animals from our previous study (Turecek and Regehr, 2019). We analyzed many thousands of boutons per animal, the distributions have been consistent across animals.

2) Authors interpret the slow IPSCs purely as a presynaptic phenomenon, i.e., asynchronous release, without considering the potential contribution of pooling and spillover, which has been shown for GABA and glycine synapses in the past.

Individual trials show that release is mediated by asynchronous quantal events (Figure 3D, Figure 4A, Figure 5A and Figure 6B). This is shown in Figure 3 of Best and Regehr, 2009, in which the timing is shown for individual quantal events. The kinetics of asynchronous release in mice are consistent with previous experiments in which individual events were detected and analyzed. We have revised the text regarding this issue in the third paragraph of the Results.

3) In Figure 6B, rDCK was labeled in red which means it receives synchronous inputs. However, no electrophysiological data support this idea (see Figure 4D). The authors need to add more data or remove the red color from rDCK.

The properties of release in the rDCK have been measured previously and were shown to be exclusively synchronous and receive input from the prepositus nucleus and not the DCN. We have clarified in the manuscript that this aspect of the summary is based on previous work and not data shown in this paper, and we have provided the appropriate citations.

4) The conclusion in the first paragraph of the Discussion: "DCN synapses are exclusively GABAergic, and lack Syt1/2, resulting in completely asynchronous release. VN synapses are mediated by both GABA and glycine, and express Syt1/2, leading to synchronized release." However, the authors showed no direct evidence between the expression of Syt1/2 and the source of inputs; all evidences are circumstantial. To make this conclusion more valid, it is necessary to check if the expression of EYFP (as a result of viral injection to DCN or VN) co-localizes with Syt1/2.

We have provided a supplementary figure (Figure 4—figure supplement 1) that shows that axons from the DCN that express YFP co-localize with Vgat, but not Syt1/2, even in a region where Syt1/2 is present in a subset of synapses. Those Syt1/2-containing inhibitory synapses lack YFP. This provides direct evidence that DCN inputs lack Syt1/2.

5) In the Materials and methods, both AAV2 and AAV9 were used, which is not consistent with the information listed in the Key Resources Table. In addition, the source of the virus is inconsistent between the table and the text.

We thank the reviewer for catching this mistake. During the course of this study, the UPenn vector core transitioned viral services to Addgene and so batches of the same virus was purchased from both vendors. We did not observe any major differences in expression level or infection efficiency between vendors. We have clarified this in the Materials and methods.

6) Please rewrite the following sentence in the Materials and methods: "Jitter was calculated by measuring the time of peak current amplitude for each individual trial was measured and calculating the standard deviation of peak times."

We have corrected the sentence and thank the reviewer for a careful reading of the manuscript.

7) Much of this work depends on the accuracy of viral expression of ChR2 in the vestibular or deep cerebellar nuclei. More quantification of YFP expression in the target regions and surrounding areas would be helpful. In Figure 3C it appears there is also high YFP expression in the cerebellar cortex, was this generally true?

We found it difficult to quantify expression of ChR2 because YFP is membrane-bound and it is often unclear whether YFP is being expressed locally or by projecting axons. For this reason we only focused on ensuring that YFP was either excluded from the DCN when the VN were targeted, or excluded from the VN and brainstem when the DCN was injected. For DCN injections, we did observe that parts of the cerebellar cortex were labeled, usually along the track of the injection pipette or superficial to the DCN itself. For VN injections, parts of the dorsal brainstem closely surrounding the VN were also sometimes labeled with YFP. We have added these details to the Materials and methods.

8) In previous work the same group has shown that synaptic responses from optogenetic stimulation of presynaptic fibers can differ from responses evoked by electrical stimulation, depending on cell-type. This raises the possibility that optogenetic stimulation affects responses from DCN terminals differently than responses from vestibular terminals, possibly accounting for the observed differences in synaptic properties, at least in part. This potential issue should be discussed briefly.

We have added a section clarifying this point. Our previous study (Jackman et al., 2014) was concerned with short term plasticity and the responses to repetitive activation. We have never seen anything to suggest that optogenetic manipulation can transform responses evoked by single stimuli from synchronous to asynchronous. In support of this, we have found that in areas where electrical stimulation evokes purely synchronous or purely asynchronous release, optical stimulation also generates purely synchronous or asynchronous release.

9) I find this study well executed. I have no concerns about the dataset, analysis or interpretation. It is intriguing that 2 different inputs use different strategies to manipulate the temporal window of inhibition. Conceptually, I find this idea fascinating and perhaps the authors could consider a short Discussion point on this.

We agree that this is an interesting topic, and this is certainly an issue that we will pursue in future studies. We have provided a few sentences on this point in the Discussion, but we would like to avoid speculation before understanding more about different IO subnuclei and how they interact with the DCN and VN.

10) Do the authors have any information on whether individual vesicles released at VN terminal onto IO cells contain both GABA and glycine? One possibility would be to assess the decay of mIPSCs. In some cases, there would be a fast monoexponential decay (glycine), in others, a slower monoexponential (GABA) and in a third population, perhaps a double exponential decay (glycine and GABA). This is more a curiosity on my part and not a requirement for the study.

Our analysis of mIPSCs suggests that for some cells a large fraction of vesicles that release glycine also release GABA (see Figure 2—figure supplement 1).

11) The Huson paper uses trains of stimuli and, here, I think things are very different. The authors could discuss this issue (single stim vs. multiple stim) and consequences for synapses that release synchronously vs. asynchronously.

The differences between single and multiple stimuli are illustrated in our new figure (Figure 7) where the two are presented side-by-side for synchronous and asynchronous synapses under near-physiological conditions.

12) The title reads awkwardly with 'nuclei' modifying synapses, almost as if the nuclei themselves are within the inferior olive. Suggest using the more grammatically correct 'nuclear' instead, or rearranging the components of the title to clarify that the synapses are from those nuclei.

We have changed the title as the reviewer suggested.